# Dense active matter model of motion patterns in confluent cell monolayers

Silke Henkes [1,2✉], Kaja Kostanjevec[3], J. Martin Collinson[3], Rastko Sknepnek [4,5✉] & Eric Bertin[6✉]

Epithelial cell monolayers show remarkable displacement and velocity correlations over distances of ten or more cell sizes that are reminiscent of supercooled liquids and active nematics. We show that many observed features can be described within the framework of dense active matter, and argue that persistent uncoordinated cell motility coupled to the collective elastic modes of the cell sheet is sufficient to produce swirl-like correlations. We obtain this result using both continuum active linear elasticity and a normal modes formalism, and validate analytical predictions with numerical simulations of two agent-based cell models, soft elastic particles and the self-propelled Voronoi model together with in-vitro experiments of confluent corneal epithelial cell sheets. Simulations and normal mode analysis perfectly match when tissue-level reorganisation occurs on times longer than the persistence time of cell motility. Our analytical model quantitatively matches measured velocity correlation functions over more than a decade with a single fitting parameter.

[1] School of Mathematics, University of Bristol, Bristol BS8 1TW, United Kingdom. [2] Institute of Complex Systems and Mathematical Biology, University of Aberdeen, Aberdeen AB24 3UE, United Kingdom. [3] School of Medicine, Medical Sciences and Nutrition, University of Aberdeen, Aberdeen AB25 2ZD, United Kingdom. [4] School of Science and Engineering, University of Dundee, Dundee DD1 4HN, United Kingdom. [5] School of Life Sciences, University of Dundee, Dundee DD1 5EH, United Kingdom. [6] Université Grenoble Alpes and CNRS, LIPHY, F-38000 Grenoble, France. ✉email: silke.henkes@bristol.ac.uk; r.sknepnek@dundee.ac.uk; eric.bertin@univ-grenoble-alpes.fr

Collective cell migration is of fundamental importance in embryonic development[1–4], organ regeneration and wound healing[5]. During embryogenesis, robust regulation of collective cell migration is key for formation of complex tissues and organs. In adult tissues, a paradigmatic model of collective cell migration is the radial migration of corneal epithelial cells across the surface of the eye[6,7]. Major advances in our understanding of collective cell migration have been obtained from in vitro experiments on epithelial cell monolayers[4,8–11]. A key observation is that collective cell migration is an emergent, strongly correlated phenomenon that cannot be understood by studying the migration of individual cells[12]. For example, forces in a monolayer are transmitted over long distances via a global tug-of-war mechanism[9]. The landscape of mechanical stresses is rugged with local stresses that are correlated over distances spanning multiple cell sizes[10]. These strong correlations lead to the tendency of individual cells to migrate along the local orientation of the maximal principal stress (plithotaxis[10,13]) and a tendency of a collection of migrating epithelial cells to move towards empty regions of space (kenotaxis[14]). Furthermore, such coordination mechanisms lead to propagating waves in confined clusters[15,16], expanding colonies[17] and in colliding monolayers[18], which all occur in the absence of inertia.

Active matter physics[19,20] offers a natural framework for describing subcellular, cellular and tissue-level processes. It studies the collective motion patterns of agents each internally able to convert energy into directed motion. In the dense limit, motility leads to a number of unexpected motion patterns, including flocking[21], oscillations[22], active liquid crystalline[20], and arrested, glassy phases[23]. In silico studies[22,24–27], in the dense regime have been instrumental in describing and classifying experimentally observed collective active motion.

Continuum active gel theories[28,29] are able to capture many aspects of cell mechanics[30], including spontaneous flow of cortical actin[31] and contractile cell traction profiles with the substrate[32]. In some cases, cell shapes form a nematic-like texture[4,33] and topological defects present in such texture have been argued to assist in the extrusion of apoptotic cells[4]. To date, however, the cell-level origin of the heterogeneity in flow patterns and stress profiles in cell sheets is still poorly understood. Many epithelial tissues show little or no local nematic order or polarization, and even where order is present, the local flow and stress patterns only follow the continuum prediction on average, while individual patterns are dominated by fluctuations. This suggests that active nematic and active gel approaches capture only part of the picture.

Confluent cell monolayers exhibit similar dynamical behaviour to supercooled liquids approaching a glass transition. One observes spatio-temporally correlated heterogeneous patterns in cell displacements[34] known as dynamic heterogeneities[35], a hallmark of the glass transition[36] between a slow, albeit flowing liquid phase and an arrested amorphous glassy state. The notion that collectives of cells reside in the vicinity of a liquid to solid transition provides profound biological insight into the mechanisms of collective cell migration. By tuning the motility and internal properties of individual cells, e.g. cell shape[37–39] or cell–cell adhesion[40], a living system can drive itself across this transition and rather accurately control cell motion within the sheet. This establishes a picture in which tissue-level patterning is not solely determined by biochemistry (e.g. the distribution of morphogens) but is also driven by mechanical cues.

In this paper, we show that the cell-level heterogeneity, that is variations in size, shape, mechanical properties or motility between individual cells of the same type inherent to any cell monolayer, together with individual, persistent, cell motility and soft elastic repulsion between neighbouring cells leads to

correlation patterns in the cell motion, with correlation lengths exceeding ten or more cell sizes. Inspired by the theory of sheared granular materials[41,42], we develop a normal modes formalism for the linear response of confluent cell sheets to active perturbations (see Fig. 1a), and derive a displacement correlation function with a characteristic length scale of flow patterns. Using numerical simulations of models for cell sheets, including a soft disk model as well as a self-propelled Voronoi model (SPV)[37,38], we show that our analytical model provides an excellent match for both types of simulations up to a point where substantial flow in the sheet begins to subtly alter the correlation functions (Fig. 1b). At the level of linear elasticity, we are able to make an analytical prediction for the velocity correlation function and the mean velocity in a generic cell sheet. We test our theoretical predictions, which apply to any confluent epithelial cell sheet on a solid substrate dominated by uncoordinated migration, with time-lapse observations of corneal epithelial cells grown to confluence on a tissue culture plastic substrate. We find very good agreement between experimental velocity correlations and analytical predictions and are, thus, able to construct fully parametrized soft disk and SPV model simulations of the system (Fig. 1c) that quantitatively match the experiment. Garcia, et al.[40] observed similar correlations and proposed a scaling theory based on coherently moving cell clusters, and either cell–substrate or cell–cell dissipation. Our approach generalises their result for cell–substrate dissipation, and we recover both the scaling results and also find quantitative agreement with the experiments presented in ref. [40].

## Results

**Model overview.** We model the monolayer as a dense packing of soft, self-propelled agents that move with overdamped dynamics, and where the main source of dissipation is cell–substrate friction. The equations of motion for cell centers are

$$\zeta \dot{\mathbf{r}}_i = \mathbf{F}_i^{\mathrm{act}} + \mathbf{F}_i^{\mathrm{int}}, \qquad (1)$$

where $\zeta$ is the cell–substrate friction coefficient, $\mathbf{F}_i^{\mathrm{act}}$ is the net motile force resulting from the cell–substrate stress transfer, and $\mathbf{F}_i^{\mathrm{int}}$ is the interaction force between cell $i$ and its neighbours. Commonly used interaction models are short-ranged pair forces with attractive and repulsive components[43] and SPV models[37,38]. Here we only require that the intercell forces can be written as the gradients of a potential energy that depends on the positions of cell centres, $\mathbf{F}_i^{\mathrm{int}} = -\nabla_{\mathbf{r}_i} V(\{\mathbf{r}_j\})$. Furthermore, we neglect cell division and extrusion for now, but we will reconsider the issue when we match simulations to experiment below. The precise form and molecular origin of the active propulsion force $\mathbf{F}_i^{\mathrm{act}}$ is a topic of ongoing debate, and interactions between cells through flocking, nematic alignment, plithotaxis and kenotaxis have all been proposed. What is clear, however, is that all alignment mechanisms occur over a substantial background of uncoordinated motility, and therefore, as a base model, we assume that the active cell forces undergo random, uncorrelated fluctuations in direction. With $\mathbf{F}_i^{\mathrm{act}} = F^{\mathrm{act}}\hat{\mathbf{n}}_i$, where $\hat{\mathbf{n}}_i$ is the unit vector that makes an angle $\theta_i$ with the $x$-axis of the laboratory frame, the angular dynamics is

$$\dot{\theta}_i = \eta_i, \qquad \langle \eta_i(t)\eta_j(t') \rangle = \frac{1}{\tau}\delta_{ij}\delta(t - t'), \qquad (2)$$

where $\tau$ sets a persistence time scale, and different cells are not coupled (Fig. 1a). This dynamics is equivalent to active Brownian particles[44], and in isolation, model cell motion is a persistent random walk. At sufficiently low driving, such models form active glasses[23,45–47], where the system moves through a series of local energy minima (i.e. spatial configurations of cells) on the time

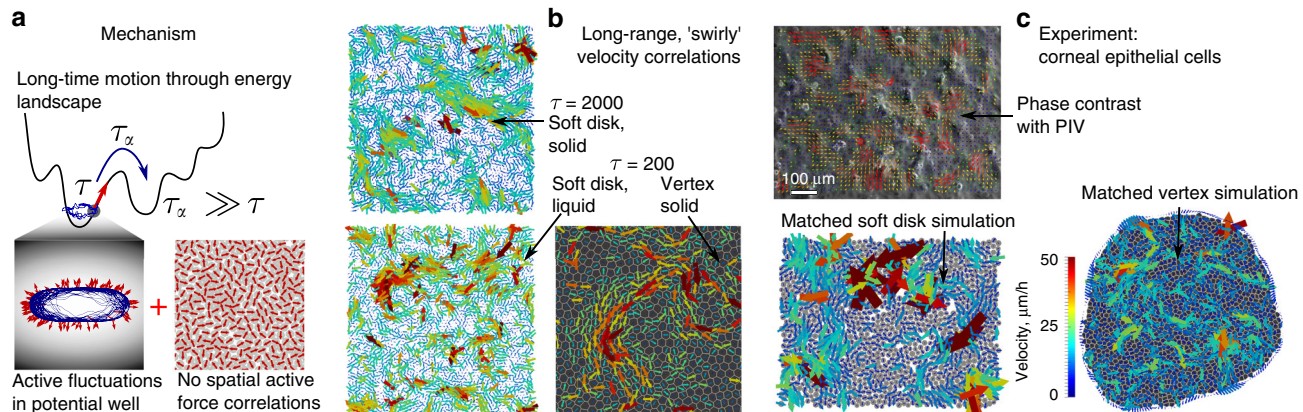

**Fig. 1 Active elasticity leads to correlated velocity fields. a** Mechanisms at the origin of the active elastic theory in the energy landscape (top), inside an energy minimum (bottom left), and between particles (bottom right). **b** Velocity fields in simulated cell sheets. Top—System-spanning correlations in a solid soft disk system at $\tau = 2000$. Bottom left—liquid soft disk system at $\tau = 200$, and bottom right—SPV model simulation at $\tau = 200$, cell outlines in white. **c** Velocity fields in experimental cell sheets. Top—sample experimental velocity field, overlaid over phase-contrast image of the cell sheet. Bottom left—particle-based best fit model to the experiment, including divisions and extrusions (visible as dark red arrows). Bottom right—SPV model-based best fit model to the experiment.

scale of the alpha-relaxation time $\tau_\alpha$, which diverges at dynamical arrest.

We now develop a linear response formalism. As shown in Fig. 1a, on time scales below $\tau_\alpha$, the self-propulsion reduces to a stochastic, time-correlated force fluctuating inside a local energy minimum. If the persistence time scale $\tau \ll \tau_\alpha$, the full dynamics can be described by a statistical average over long periods fluctuating around different energy minima, by assuming that the brief periods during which the system rearranges do not contribute appreciably (see also ref. [47]). We linearize the interaction forces in the vicinity of an energy minimum, i.e. a mechanically stable or jammed configuration $\{\mathbf{r}_i^0\}$ by introducing $\delta\mathbf{r}_i = \mathbf{r}_i - \mathbf{r}_i^0$. After introducing the active velocity $v_0 = F^{act}/\zeta$, Eq. (1) becomes

$$\zeta\delta\dot{\mathbf{r}}_i = \zeta v_0 \hat{\mathbf{n}}_i - \sum_j \mathbf{K}_{ij} \cdot \delta\mathbf{r}_j, \qquad (3)$$

where $\mathbf{K}_{ij} = \frac{\partial^2 V(\{\mathbf{r}_i\})}{\partial\mathbf{r}_i\partial\mathbf{r}_j}|_{\{\mathbf{r}_j^0\}}$ is the dynamical matrix[48], organised as $2 \times 2$ blocks corresponding to cells $i$ and $j$. In this limit, we can solve the dynamics exactly, see Supplementary Note 1.

**Normal mode formulation.** Assuming that there are a sufficient number of intercell forces to constrain the tissue to be elastic at short time scales, the dynamical matrix has $2N$ independent normal modes $\boldsymbol{\xi}^\nu$ with positive eigenvalues $\lambda_\nu$. If we project Eq. (3) onto the normal modes, we obtain

$$\zeta\dot{a}_\nu = -\lambda_\nu a_\nu + \eta_\nu, \qquad (4)$$

where $a_\nu = \sum_i \delta\mathbf{r}_i \cdot \boldsymbol{\xi}_i^\nu$ and the self-propulsion force has been projected onto the modes, $\eta_\nu = \zeta v_0 \sum_i \hat{\mathbf{n}}_i \cdot \boldsymbol{\xi}_i^\nu$. The self-propulsion then acts like a time-correlated Ornstein-Uhlenbeck noise (see Supplementary Note 1), with $\langle\eta_\nu(t)\eta_{\nu'}(t')\rangle = \frac{1}{2}\zeta^2 v_0^2 \exp(-|t-t'|/\tau)\delta_{\nu,\nu'}$. We can integrate Eq. (4) and obtain the moments of $a_\nu$. In particular, the mean energy per mode is given by

$$E_\nu = \frac{1}{2}\lambda_\nu\langle a_\nu^2\rangle = \frac{\zeta v_0^2\tau}{4(1+\lambda_\nu\tau/\zeta)}, \qquad (5)$$

explicitly showing that equipartition is broken due to the mode-dependence induced by $\lambda_\nu$ in Eq. (5). In the limit $\tau \to 0$, we recover an effective thermal equilibrium, $E_\nu \to \zeta v_0^2\tau/4 := T_{\text{eff}}/2$,

where $T_{\text{eff}} = \zeta v_0^2\tau/2$, consistent with previous work[37,45]. In Fig. 2a, the leftmost column is for a simulated thermal system, with properties that are nearly indistinguishable from the $\tau = 0.2$ results. In the opposite, high persistence limit when $\tau \to \infty$, we obtain instead $E_\nu = \zeta^2 v_0^2/4\lambda_\nu$, i.e. a divergence of the contribution of the lowest modes. A predominance of the lowest modes in active driven systems was also noted in ref. [22,37,49].

It thus becomes clear that for large values of $\tau$, $T_{\text{eff}}$ can no longer be interpreted as temperature since the fluctuation-dissipation theorem is no longer valid. An analogous result to the $\tau \to \infty$ limit has been obtained in granular material with an externally applied shear[41,42], showing that the mechanisms at play are generic, and that tuning $\tau$ allows active systems to bridge between features of thermal systems and (self-)sheared systems (see also ref. [37,47]). However, the glass transition lies on a curve of constant $T_{\text{eff}}$ with moderate $\tau$ contributions (Fig. 2 and ref. [46]), making $T_{\text{eff}}$ a convenient parameter, in spite of its lack of genuine thermodynamic interpretation.

In order to make connections to experiments on cells sheets, we compute several directly measurable quantities. One measure that is easily extracted from microscopy images is the velocity field, using particle image velocimetry (PIV)[50]. We compute the Fourier space velocity correlation function, $\langle|\mathbf{v}(\mathbf{q})|^2\rangle = \langle\mathbf{v}(\mathbf{q}) \cdot \mathbf{v}^*(\mathbf{q})\rangle$, with $\mathbf{v}(\mathbf{q}) = 1/N\sum_{j=1}^N e^{i\mathbf{q}\cdot\mathbf{r}_j^0}\delta\dot{\mathbf{r}}_j$, where the $\{\mathbf{r}_j^0\}$ are the positions of the cell centres at mechanical equilibrium. Expanding over the normal modes, and taking into account the statistical independence of the time derivatives $\dot{a}_\nu$ of the modes amplitudes for different modes, we first derive (see Supplementary Note 1) the mode correlations $\langle\dot{a}_\nu^2\rangle = v_0^2/[2(1+\lambda_\nu\tau/\zeta)]$, so that in Fourier space, we obtain

$$\langle|\mathbf{v}(\mathbf{q})|^2\rangle = \sum_\nu \frac{v_0^2}{2(1+\lambda_\nu\tau/\zeta)} |\boldsymbol{\xi}_\nu(\mathbf{q})|^2, \qquad (6)$$

where $\boldsymbol{\xi}_\nu(\mathbf{q})$ is the Fourier transform of the vector $\boldsymbol{\xi}_i^\nu$.

**Continuum elastic formulation.** In most practical situations, it is impossible to extract either the normal modes or their eigenvalues. While it is possible to do so in, e.g. colloidal particle experiments[51,52], the current methods are strictly restricted to thermal equilibrium, and also require an extreme amount of data. Fortunately, the results above are easily recast into the language of

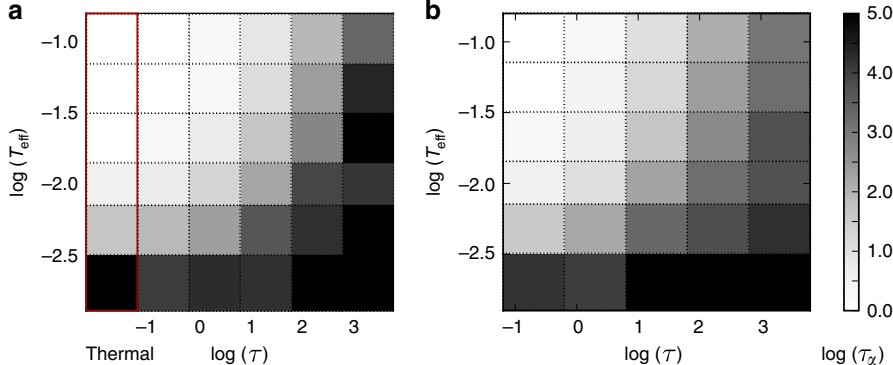

**Fig. 2 Glassy dynamics.** Alpha-relaxation time $\tau_\alpha$ as a function of the persistence time $\tau$ and effective temperature $T_{\rm eff} = \zeta v_0^2 \tau/2$. The gray scale indicates $\log \tau_\alpha$. **a** Soft disk model at $\phi = 1$, the leftmost column is for a thermal system at $T = T_{\rm eff}$. **b** SPV model at $\bar{p}_0 = 3.6$ (see Methods).

solid state physics[53]. We rewrite Eq. (3) as

$$\zeta \dot{\mathbf{u}}(\mathbf{R}) = \zeta v_0 \hat{\mathbf{n}}(\mathbf{R}) - \sum_{\mathbf{R}'} \mathbf{D}(\mathbf{R} - \mathbf{R}')\mathbf{u}(\mathbf{R}'), \qquad (7)$$

where $\mathbf{u}(\mathbf{R})$ denotes the elastic deformations from the equilibrium positions $\mathbf{R}$ in the solid, and $\mathbf{D}(\mathbf{R} - \mathbf{R}')$ is the continuum dynamical matrix. The normal modes of the system are now simply Fourier modes with

$$-i\zeta\omega\mathbf{u}(\mathbf{q}, \omega) = \mathbf{F}^{\rm act}(\mathbf{q}, \omega) - \mathbf{D}(\mathbf{q})\mathbf{u}(\mathbf{q}, \omega), \qquad (8)$$

where $\mathbf{F}^{\rm act}(\mathbf{q}, \omega) = \zeta v_0 \int_{-\infty}^{\infty} dt \sum_{\mathbf{R}} \hat{\mathbf{n}}(\mathbf{R}, t) e^{i\omega t} e^{i\mathbf{q}\cdot\mathbf{R}}$ and $\mathbf{D}(\mathbf{q})$ are Fourier transforms of the active force and the dynamical matrix, respectively. Note that we assume that the system has a finite volume, so that Fourier modes are discrete. At scales above the cell size $a$, the noise $\hat{\mathbf{n}}(\mathbf{R}, t)$ is spatially uncorrelated, and we find the noise correlators (see Supplementary Note 2)

$$\langle \mathbf{F}^{\rm act}(\mathbf{q}, \omega) \cdot \mathbf{F}^{\rm act}(-\mathbf{q}, \omega') \rangle = 2\pi N \zeta^2 v_0^2 \frac{2\tau}{1 + (\tau\omega)^2} \; \delta(\omega + \omega'). \qquad (9)$$

The dynamical matrix $\mathbf{D}(\mathbf{q})$ has two independent eigenmodes in two dimensions, one longitudinal $\hat{\mathbf{q}}$ with eigenvalue $B + \mu$ and one transverse one $\hat{\mathbf{q}}^{\perp}$ with eigenvalue $\mu$, where $B$ and $\mu$ are the bulk and shear moduli, respectively. We can then decompose our solution into longitudinal and transverse parts, $\mathbf{u}(\mathbf{q}, \omega) = u_{\rm L}(\mathbf{q}, \omega)\hat{\mathbf{q}} + u_{\rm T}(\mathbf{q}, \omega)\hat{\mathbf{q}}^{\perp}$. We are interested in the equal time, Fourier transform of the velocity, which we find to be (see Supplementary Note 2)

$$\langle |\mathbf{v}(\mathbf{q})|^2 \rangle = \frac{N v_0^2}{2} \left[ \frac{1}{1 + (\xi_{\rm L}q)^2} + \frac{1}{1 + (\xi_{\rm T}q)^2} \right], \qquad (10)$$

where we have introduced the longitudinal and transverse correlation lengths $\xi_{\rm L}^2 = (B + \mu)\tau/\zeta$ and $\xi_{\rm T}^2 = \mu\tau/\zeta$. Note that there are subtle differences in prefactors between expressions for velocity correlation function in Fourier space (cf., Eq. (6), Eq. (10) and Eq. (52) in Supplementary Note 2), and that in two dimensions, $[\mu/\zeta] = [B/\zeta] = L^2 T^{-1}$. As discussed in detail in Supplementary Note 2, these differences are due the use of discrete vs. continuum Fourier transforms and are important for comparison with simulations and experiments. Finally, the mean-square velocity of the particles $\langle |\mathbf{v}|^2 \rangle = \langle \frac{1}{N} \sum_i |\mathbf{v}_i|^2 \rangle$ decreases with active correlation time as

$$\langle |\mathbf{v}|^2 \rangle = \frac{v_0^2 a^2}{8\pi} \left[ \frac{1}{\xi_{\rm L}^2} \log\left(1 + \xi_{\rm L}^2 q_{\rm m}^2\right) + \frac{1}{\xi_{\rm T}^2} \log\left(1 + \xi_{\rm T}^2 q_{\rm m}^2\right) \right], \quad (11)$$

where $q_{\rm m} = 2\pi/a$ is the maximum wavenumber and the high-$q$

cutoff $a$ is of the order of the cell size. Eq. (10) shows that the correlation length of the system scales as $\sqrt{\tau}$. In the limit $\tau \to \infty$, $\langle |\mathbf{v}(\mathbf{q})|^2 \rangle$ diverges at low $q$, as was found in ref. [54]. The dominant scaling $\langle |\mathbf{v}|^2 \rangle \sim 1/\xi^2$ is the same as results from the scaling Ansatz for cell–substrate dominated coordinated motion obtained by ref. [40].

While Eq. (10) is elegant, correlations of cell velocities expressed in the Fourier space are not easy to interpret. Therefore, we derive a more intuitive, real-space expression for the correlation of velocities of cells separated by $r$, defined as

$$C_{vv}(\mathbf{r}) = \frac{1}{L^2} \int d^2 \mathbf{r}_0 \langle \mathbf{v}(\mathbf{r}_0 + \mathbf{r}) \cdot \mathbf{v}(\mathbf{r}_0) \rangle. \qquad (12)$$

In the infinite size limit $L \to \infty$, the real-space correlation function $C_{vv}(\mathbf{r})$ can be evaluated from the Fourier correlation $\langle |\mathbf{v}(\mathbf{q})|^2 \rangle$ as

$$C_{vv}(\mathbf{r}) = \frac{a^2}{(2\pi)^2 N} \int d^2 \mathbf{q} \; \langle |\mathbf{v}(\mathbf{q})|^2 \rangle \; e^{-i\mathbf{q}\cdot\mathbf{r}}. \qquad (13)$$

Using Eq. (11), one finds the explicit result (see Supplementary Note 2)

$$C_{vv}(\mathbf{r}) = \frac{a^2 v_0^2}{4\pi} \left[ \frac{K_0(r/\xi_{\rm L})}{\xi_{\rm L}^2} + \frac{K_0(r/\xi_{\rm T})}{\xi_{\rm T}^2} \right], \qquad (14)$$

where $K_0$ is the modified Bessel function of the second kind. Note that this expression describes velocity correlations for $r > a$, with $a$ the cell size. For $r/\xi_{\rm L,T} \gg 1$, i.e. for distance much larger than the correlation lengths,

$$C_{vv}(\mathbf{r}) \approx \frac{a^2 v_0^2}{4\pi} \sqrt{\frac{\pi}{2r}} \left( \frac{e^{-r/\xi_{\rm L}}}{\xi_{\rm L}^{3/2}} + \frac{e^{-r/\xi_{\rm T}}}{\xi_{\rm T}^{3/2}} \right), \qquad (15)$$

i.e. as expected and consistent with the results of ref. [40], $C_{vv}$ decays exponentially at large distances.

**Comparison to simulations.** We proceed to compare predictions made in the previous section to the correlation function measured in numerical simulations of an active Brownian soft disk model, as well as to an SPV model. The active Brownian model is defined by Eq. (1) and Eq. (2), with self-propulsion force $\mathbf{F}_i^{\rm act} = v_0 \hat{\mathbf{n}}_i$ and pair interaction forces $\mathbf{F}_{ij}$ that are purely repulsive. We simulate a confluent sheet in this model by setting the packing fraction to $\phi = 1$ in periodic boundary conditions. The SPV model is the same as introduced in refs. [37,38], and assumes that every cell is defined by the Voronoi tile corresponding to its centre. For this model, we choose the dimensionless shape factor $\bar{p}_0 = 3.6$, putting the passive system into the solid part of the phase

diagram[37,55], and we employ open boundary conditions. Please see the method section for full details of the numerical models and simulation protocols.

The effective temperature $T_{\mathrm{eff}} = \zeta v_0^2 \tau / 2$ has emerged as a good predictor of the active glass transition[45,54], at least at low $\tau$, and we use it together with $\tau$ itself as the axes of our phase diagram. The liquid or glassy behaviour of the model can be characterised by the alpha-relaxation time $\tau_\alpha$. Fig. 2 provides a coarse-grained phase diagram where $\tau_\alpha$ is represented in grey scale as a function of persistence time $\tau$ and $T_{\mathrm{eff}}$. For a fixed persistence time, the system is liquid at high enough temperature and glassy at low temperature, as expected. Now fixing the effective temperature, the system becomes more glassy when $\tau$ increases. This non-trivial result is consistent with the recent RFOT theory of the active glass transition[46] and related simulation results[46,47]. It can be partly understood from the fact that $v_0$ decreases when $\tau$ increases at fixed $T_{\mathrm{eff}}$, meaning that the active force decreases and it becomes more difficult to cross energy barriers. In contrast, existing mode coupling theories of the active glass transition[54] only apply in the small $\tau$ regime. We note that the features of the active glass transition of the soft disk model and the SPV model are very similar.

As is apparent from Fig. 1b, the growing correlation length with increasing $\tau$ is readily apparent as swirl-like motion (see also Supplementary Movies 1-4). Fig. 3a-c shows the Fourier velocity correlation $\langle |\mathbf{v}(\mathbf{q})|^2 \rangle$ measured in the numerical simulations for different values of $v_0$, after normalizing $\langle |\mathbf{v}(\mathbf{q})|^2 \rangle$ by $v_0^2 N$. In panel A, we show that for soft disks at low $T_{\mathrm{eff}} = 0.005$, where the system is solid, the correlation function develops a dramatic $1/q^2$ slope as $\tau$ increases (dots), exactly in line with our modes predictions (lines). We can determine the bulk and shear moduli of the soft disk system ($B = 1.684 \pm 0.008$, $\mu = 0.510 \pm 0.004$, see Methods section) and then draw the predictions of Eq. (10) on the same plot (dashed lines). At low $q$, where the continuum elastic approximation is valid, we have excellent agreement, and at larger $q$, the peak associated with the static structure factor becomes apparent (in the limit $\tau \to 0$, the correlation function reduces to $S(q)$, see Supplementary Fig. 2). In panel C, we show the same simulation results for the SPV model (dots), accompanied by the continuum predictions (dashed lines) using $B = 7.0$ and $\mu = 0.5$, as estimated from ref. [55] for $\bar{p}_0 = 3.6$. We did not compute normal modes for the SPV model. Note that due to $\mu \ll \mu + B$, the contribution of the transverse correlations dominate the analytical results in both cases. In panel B, we show the soft disk simulation at $\tau = 20$ when the transition to a liquid is crossed as a function of $T_{\mathrm{eff}}$. Deviations from the normal mode predictions become apparent only at the two largest values of $T_{\mathrm{eff}}$ when $\tau > \tau_\alpha$ (Fig. 2a), and even in these very liquid systems, a significant activity-induced correlation length persists. In Supplementary Fig. 3, we show that for all $\tau$ and both soft disk and SPV models, our predictions remain in excellent agreement with the simulations for $T_{\mathrm{eff}} = 0.02$, where $\tau \lesssim \tau_\alpha$. In Fig. 3d, we show the mean-square velocity normalized by $v_0$ as a function of the dimensionless transverse correlation length $\xi_T / a \sim \sqrt{\tau}$, for all our simulations, using $a = \sigma$, the particle radius. The dramatic drop corresponds to elastic energy being stored in distortions of the sheet, and it is in very good agreement with our analytical prediction in Eq. (11) (solid lines). In Supplementary Fig. 4, we compare our numerical results for the spatial velocity correlations to the analytical prediction Eq. (14). The data and the predictions are in reasonable agreement.

## Comparison to experiment

We now compare our theoretical predictions and numerical simulations with experimental data obtained from immortalised human corneal epithelial cells grown on a tissue culture plastic substrate (see Methods section and Supplementary Movie 5). We use PIV to extract the velocity fields corresponding to collective cell migration (Fig. 4a, Fig. 1c and Supplementary Movie 6). We first extract a mean velocity of $\bar{v} = \sqrt{\langle |\mathbf{v}(\mathbf{r},t)|^2 \rangle} = 12 \pm 2 \mu m/h$ ($n = 5$ experiments, see Fig. 4e), consistent with the typical mean velocities of confluent epithelial cell lines grown on hard substrates. To reduce the effects of varying mean cell speed at different times and in different experiments, we use $\bar{\mathbf{v}}(\mathbf{r},t) = \mathbf{v}(\mathbf{r},t) / \sqrt{\langle |\mathbf{v}(\mathbf{r},t)|^2 \rangle_{\mathbf{r}}}$, i.e., the velocity normalized by its mean-square spatial average at that moment in time. Using direct counting, we find an area per cell of $\langle A \rangle \approx 380 \mu m^2$ corresponding to a particle radius of $\langle \sigma \rangle \approx 11 \mu m$, setting the microscopic length scale $a$. To compare the experimental result to our theoretical predictions, we perform a Fourier transform on the PIV velocity field and compute $\langle |\bar{\mathbf{v}}(\mathbf{q})|^2 \rangle$, shown in Fig. 4b. Using the results from Eq. (11) we can rewrite Eq. (10) as

$$\langle |\bar{\mathbf{v}}(\mathbf{q})|^2 \rangle = \frac{N}{2} \left( \frac{v_0}{\bar{v}} \right)^2 \left[ \frac{1}{1 + \xi_L^2 q^2} + \frac{1}{1 + \xi_T^2 q^2} \right], \quad (16)$$

where $\xi_L^2$ and $\xi_T^2$ are the longitudinal and transverse squared correlation lengths (with units of $\mu m^2$) defined below Eq. (10). As can be seen from Eq. (11), the ratio $v_0 / \bar{v}$ is a function only of the dimensionless ratios $\xi_L / a$ and $\xi_T / a$. If we further make the plausible assumption that the ratio of elastic moduli is the same as in the soft disk simulations, $(\mu + B)/\mu = 4.3$, the correlation lengths $\xi_L$ and $\xi_T$ are not independent. Therefore, we are left with a single fitting parameter, $\xi_T$. The best fit to the theory is obtained with $\xi_T = 100$ μm as indicated by the solid black line in Fig. 4a, with the interval of confidence denoted by dashed lines. The $q = 0$ intercept of the correlation function gives a ratio $v_0 / \bar{v} \approx 10$, corresponding to the high activity limit where most self-propulsion is absorbed by the elastic deformation of the cells. Consistent with this, on the dimensionless plot Fig. 3d we are located at the point $\xi_T / a \approx 5$, $\langle v \rangle / v_0 \approx 0.1$, on the right, strongly active side. The deviations between theory and experiment in the tail of the distribution are not due to loss of high-$q$ information in imaging, as far as we could determine, and the disappearance of the peak at high $q$ is particularly striking in this context.

In Fig. 4c, we show the real-space velocity correlations for the experiments, and the analytical prediction Eq. (14) with $\xi_T = 100$ μm. We obtain a very good fit for experiments 1, 2, 3 and 7, but experiments 5 and 6 have significantly longer-ranged correlations. This indicates that the precise value of the correlation length is very sensitive to the exact experimental conditions that are not simple to accurately control. The qualitative features of the correlation function are, however, robust. Note that experiments 5 and 6 have the same mean density as experiments $1 - 3$, 7.

To match experiments and simulations, we consider the temporal autocorrelation function $\langle \mathbf{v}(t) \cdot \mathbf{v}(0) \rangle$ in Fig. 4d. As Eq. (57) in Supplementary Note 2 shows, it is a complex function with a characteristic inverse S-shape that also depends on the moduli and $q_{\mathrm{m}}$. Using the value of $\xi_T^2 = 10^4 \mu m^2$ extracted from fitting $\langle |\bar{\mathbf{v}}(\mathbf{q})|^2 \rangle$ and the ratio $(\mu + B)/\mu = 4.3$, we used different $\mu/\zeta$ and $\tau$ compatible with $\xi_T^2 = \mu \sigma^2 / \zeta \tau$ to obtain the best (numerically integrated) analytical fit to the experimental result. We settled on a best fit autocorrelation time of $\tau = 2.5$ h and $\mu/(\sigma^2 \zeta) = 60.5 \, h^{-1}$ (black line in Fig. 4d). Experiments 5 and 6 have significantly longer autocorrelation times, and we achieve a good fit to experiment 5 for $\tau_5 = 20$ h and the same $\mu/\zeta$ (light gray line). This is consistent with the longer spatial correlations observed in Fig. 4d, where the light grey line corresponds to $\xi_T = 100$ μm $\sqrt{\tau_5/\tau} \approx 283 \mu$ m. There is also potentially weak local cell alignment in the experiment, not considered in the present theory.

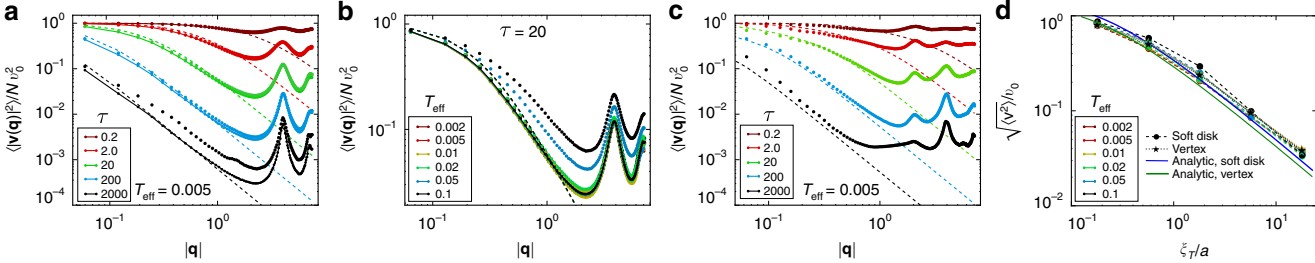

**Fig. 3 Behaviour of velocity correlation functions. a** Velocity correlations in Fourier space for the soft disk model for different $\tau$ at $T_{\text{eff}} = 0.005$ in the glassy phase and **b**: for changing $T_{\text{eff}}$ at $\tau = 20$. Dots correspond to the simulated velocity correlation function, solid lines are the results from the normal mode expansion Eq. (6), and the dashed lines are the continuum elasticity predictions Eq. (10). **c** Fourier velocity correlations for the SPV model, together with analytical prediction. **d** Mean-square velocity as a function of $\xi_T/a$ for both soft disks and the SPV model and analytical predictions (solid lines). All correlation functions have been normalized by $v_0^2 = 2T_{\text{eff}}/\tau$, and there are no fit parameters in the predictions.

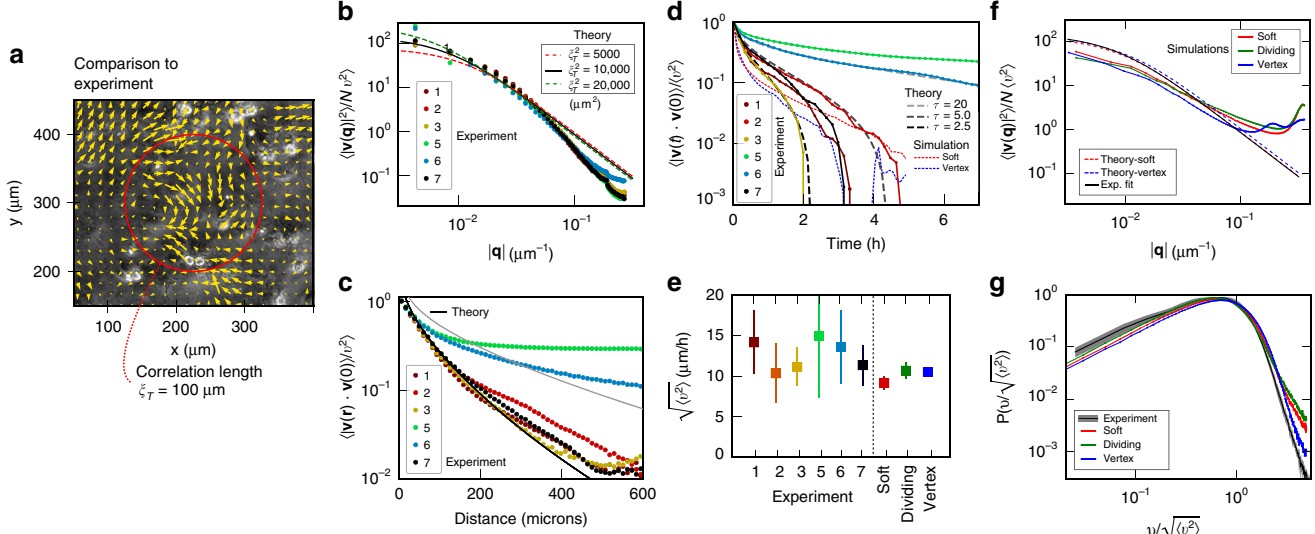

**Fig. 4 Experimental results and comparison to theory and simulations. a** Cell sheet image overlaid with PIV arrows, and best fit length scale $\xi_T$. **b** Experimental Fourier velocity correlation function normalised by mean velocity (dots), and best fit to theory with a single stiffness parameter $\xi_T^2 = \mu\tau/\zeta$. **c** Experimental real-space velocity correlation function (dots) and theoretical prediction. **d** Velocity autocorrelation function for the experiments, theoretical predictions for different $\tau$, and simulation results for the same soft disk and SPV model simulations as in **f**. **e** Mean velocity magnitude for the experiments, and the soft disk, dividing soft disk and SPV model simulations. **f** Numerical results for soft, soft dividing, and SPV model interactions (solid lines), and theoretical predictions (dashed lines) for the same $\tau = 2.5h$, $k = 55\mu_m^{-2}h^{-1}$ and $v_0 = 90\,\mu mh^{-1}$ (see text). **g** Normalized velocity distribution from experiment (black with grey confidence interval), and the three simulated models.

We can now fully parametrise particle and SPV model simulations to the experiment as follows. Our results for $\bar{v}$ and the ratio $v_0/\bar{v}$ can be combined to give an initial estimate of $v_0 = 120\,\mu m\,h^{-1}$. Then, the normalised time autocorrelation function of the cell velocities is only a function of $\xi$ and $\tau$, and we can use it to determine the elastic moduli. Then, finally, we can determine the appropriate model parameters: In Fig. 4b, the red and blue dashed lines show Eq. (10) with $B$ and $\mu$ chosen with the same ratio as in the previous particle (respectively, SPV) simulations. From these values, we can approximate the parameter values $k/\zeta = \mu/\sigma^2$ for the particle model and $K/\zeta = \mu/\langle A\rangle^2$, $\Gamma/\zeta = \mu/\langle A\rangle$ for the SPV model. The solid red and blue curves in Fig. 4b show the best fit simulations that we obtain this way, for $k/\zeta = \Gamma/\zeta = 55\,h^{-1}$, $K/\zeta = 0.454\,\mu m^{-2}\,h^{-1}$ and $v_0 = 90\,\mu m\,h^{-1}$, and snapshots are shown in Fig. 1c (see also Supplementary Movies 7, 9 and 10). The red and blue dashed lines in Fig. 4d show the autocorrelations of our matched simulations for soft disk and vertex simulations, respectively.

Our results are in quantitative agreement with ref. [40] for confluent but still motile cells, with a reported maximum correlation length of $\xi = 100\,\mu m$, and a cell crawling speed that drops by a factor of 10 from $\sim 90\,\mu m\,h^{-1}$ at low density to this point of maximal correlation. Note also that we have an elastic time scale $(k/\zeta)^{-1} \approx 0.02h$, much shorter than our correlation time scale $\tau = 2.5h$, confirming again that we are in the strongly active regime.

As can be seen in Supplementary Movie 5, a significant number of divisions take place in the epithelial sheet during the 48h of the experiment. While it is difficult to adapt our theory to include divisions, we can simulate our particle model with a steady-state division and extrusion rate at confluence using the model developed in refs. [56,57]. With a typical cell cycle time of $\tau_{\text{div}} = 48h$, we obtain results (green line) that are very similar to the model without division (red line), suggesting that typical cell division rates do not change the velocity correlations noticeably (see also Supplementary Movie 8). This result is consistent with

the observed separation of motility time scale $\tau$ and division time scale $\tau_{div}$. We have also considered the effect of weak polar alignment between cells, using the model from ref. [22]. For weak alignment with time scales $\tau_\nu \geq \tau$ that do not lead to global flocking of the sheet, which we do not observe, $\langle |\mathbf{v}(\mathbf{q})|^2 \rangle$ does not change significantly, though we find somewhat longer autocorrelation times. Finally, the simulations can also give us information about the velocity distribution function, a quantity that is not accessible from our theory. In Fig. 4d, we show the experimental normalised velocity distribution (black line with grey confidence interval), together with the distribution we find from the best fit simulations (coloured lines). As can be seen, there is an excellent match in particular with the SPV model simulation. The particle model with division has additional weight in the tail due to the particular division algorithm implemented in the model (overlapping cells pushing away from each other).

## Discussion

In this study, we have developed a general theory of motion in dense epithelial cell sheets (or indeed other dense active assemblies[58]) that only relies on the interplay between persistent active driving and elastic response. We find an emerging correlation length that depends only on elastic moduli, the substrate friction coefficient and scales with persistence time as $\tau^{1/2}$. While we found an excellent match between theory and simulations, further experimental validations with different cell lines and on larger systems should be performed. Note that without a substrate, the mechanisms of cell activity are very different[59]. More generally, including cell–cell dissipation in addition to cell–substrate dissipation could significantly modify scalings[40], a known result in continuum models of dry (substrate dissipation) vs. wet (internal dissipation) active materials. Due to the suppression of tangential slipping between cells, it could also be responsible for the disappearance of the high-$q$ peak in the velocity correlations. The assumption of uncoordinated activity between cells is a strong one, and it will be interesting to extend the theory by including different local mechanisms of alignment[22,49]. From a fundamental point of view, our theoretical results (and also the results of ref. [22]) are examples of a larger class of non-equilibrium steady-states that can be treated using a linear response formalism[60].

## Methods

**Experiment.** Spontaneously immortalised, human corneal epithelial cells (HCE-S) (Notara & Daniels, 2010) were plated into a 12-well plate using growth medium consisting of DMEM/F12 (Gibco) Glutamax, 10% fetal bovine serum, and 1% penicillin/streptomycin solution (Gibco). The medium was warmed to 37°C prior to plating and the cells were kept in a humidified incubator at 37 °C and an atmosphere of 5% $CO_2$ overnight until the cells reached confluence. Before imaging the cells were washed with PBS and the medium was replaced with fresh medium buffered with HEPES. The cells were imaged using a phase-contrast Leica DM IRB inverted microscope enclosed in a chamber which kept the temperature at 37 °C. The automated time-lapse imaging setup took an image at 10 minute intervals at a magnification of ×10, corresponding to a field of view of 867 × 662 µm that was saved at a resolution of 1300 × 1000 pixels. The total experimental run time for each culture averaged 48 h, or 288 separate images. The collected data consists of seven experimental imaging runs, of which number 3 and 4 were consecutive on the same well-plate (number 4 was not used in this article). Cell extrusions were counted at three time points during the experimental run by direct observation and counting from the still image (extruded cells detach from the surface and round up, appearing as white circles above the cell sheet in phase contrast). From this data, a typical cell number of $N = 1400$, and a typical cell radius of $r = 10.95$ µm were extracted. Cell movements were determined using Particle Image Velocimetry (PIV), using an iterative plugin for ImageJ (https://sites.google.com/site/qingzongtseng/piv). At the finest resolution (level 3), it provides displacement vectors on a 54 × 40 grid, corresponding to a resolution of 16 µm in the $x$ and the $y$-direction, i.e. slightly less than 1 cell diameter. The numerous extruded cells and the nucleoli inside the nuclei acted as convenient tracer particles for the PIV allowing for accurate measurements.

**Simulations.** The main simulations consist of $N = 3183$ particles simulated with either soft repulsion, or the SPV model (in literature also refered to as the Active Vertex Model (AVM)) potential, using SAMoS (https://github.com/sknepneklab/SAMoS). The interaction potential for soft harmonic disks is $V_i = \sum_j \frac{k}{2}(\sigma_i + \sigma_j - |\mathbf{r}_j - \mathbf{r}_i|)^2$ if $|\mathbf{r}_j - \mathbf{r}_i| \leq \sigma_i + \sigma_j$ and 0 otherwise. To emulate a confluent cell sheet, we used periodic boundary conditions at packing fraction $\phi = 1$, where $\phi = \sum_i \pi \sigma_i^2 / L^2$ and thus double-counts overlaps. We also introduce 30% polydispersity in radius, to emulate cell size heterogeneity. At this density, the model at zero activity is deep within the jammed region ($\phi > 0.842$) and has a significant range of linear response. The disk simulations fitted to experiment also include a short-range attractive region as in[57] with $\varepsilon = 0.15$.

For the SPV, cells are defined as Voronoi polygonal tiles around cell centers, and the multiparticle interaction potential is given by $V_i = \frac{K}{2}(A_i - A_0)^2 + \frac{\Gamma}{2}(P_i - P_0)^2$, where $A_i$ is the area of the tile, and $P_i$ is its perimeter, $K$ and $\Gamma$ are the area and perimeter stiffness coefficients and $A_0$ and $P_0$ are reference area and perimeter, respectively. SPV is confluent by construction, and its effective rigidity is set by the dimensionless shape parameter $\bar{p}_0 = P_0 / \sqrt{A_0}$, with a transition from a solid to a fluid that occurs for $\bar{p}_0 \approx 3.812$. We simulate the model at $\bar{p}_0 = 3.6$, well within the solid region at zero activity[37,55], and also introduce 30% variability in $A_0$. AVM was implemented with open boundary conditions, and we use a boundary line tension $\lambda = 0.3$ to avoid a fingering instability at the border that appears especially at large $\tau$.

Both models are simulated with overdamped active Brownian dynamics $\zeta \dot{\mathbf{r}}_i = v_0 \hat{\mathbf{n}}_i - \nabla_{\mathbf{r}_i} V_i$, where the orientation vector $\hat{\mathbf{n}}_i = (\cos \theta_i, \sin \theta_i)$ follows $\dot{\theta}_i = \eta_i$, $\langle \eta_i(t) \eta_j(t') \rangle = \frac{1}{\tau} \delta_{ij} \delta(t - t')$. Equations of motions are integrated using a first order scheme with time step $\delta t = 0.01$. Simulations are $5 \times 10^4$ time units long, with snapshots saved every 50 time units, and the first 1250 time units of data are discarded in the data analysis.

**Velocity correlations and glassy dynamics.** We compute the velocity correlation function for a given simulation directly from particle positions and velocities by first computing the Fourier transform. Then for a given $\mathbf{q}$ and configuration, the correlation function is $|\mathbf{v}(\mathbf{q})|^2 = \mathbf{v}(\mathbf{q}) \cdot \mathbf{v}^*(\mathbf{q})$, of which we then take a radial $\mathbf{q}$ average, followed by a time average. The procedure is identical for the experimental PIV fields using the grid positions and velocities, with $N = 54 \times 40$ grid points as normalization. We compute the $\alpha$-relaxation time from the self-intermediate scattering function $S(q, t) = \langle \frac{1}{N} \sum_j e^{i\mathbf{q} \cdot (\mathbf{r}_j(t_0 + t) - \mathbf{r}_j(t_0))} \rangle_{t_0, |\mathbf{q}| = q}$, where the angle brackets indicate time and radial averages. At $q = 2\pi/\sigma$, we determine $\tau_\alpha$ as the first time point where $S(q, t) < 0.5$, bounded from above by the simulation time.

**Normal mode analysis.** The normal modes are the eigenvalues and eigenvectors of the Hessian matrix $\mathbf{K}_{ij} = \partial^2 V(\{\mathbf{r}_i\}) / \partial \mathbf{r}_i \partial \mathbf{r}_j$, evaluated at mechanical equilibrium. We first equilibrate the $t = 2500$ snapshot with $v_0 = 0$ for $2 \times 10^5$ time steps, equivalent to a steepest descent energy minimization. We made sure that results are not sensitive to the choice of snapshot as equilibration starting point (with the exception of the deviations apparent in Fig. 3 at $\tau = 2000$). For the soft disk model, each individual $ij$ contact with contact normal $\hat{\mathbf{n}}_{ij}$ and tangential $\hat{\mathbf{t}}_{ij}$ vectors contributes a term $K_{ij} = -k \hat{\mathbf{n}}_{ij} \times \hat{\mathbf{n}}_{ij} + |\mathbf{f}_{ij}| \hat{\mathbf{t}}_{ij} \times \hat{\mathbf{t}}_{ij}$ to the $ij$ $2 \times 2$ off-diagonal element of the matrix and $-K_{ij}$ is added to the $ii$ diagonal element[48]. We use the NumPy eigh function (numpy.linalg.eigh). As the system is deep in the jammed phase, with the exception of two translation modes, all eigenvalues of the Hessian are positive. We compute the Fourier spectrum of mode $\nu$ through $\xi_\nu(\mathbf{q}) = \frac{1}{N} \sum_j e^{i\mathbf{q} \cdot \mathbf{r}_j} \xi_\nu^j$ and then $|\xi_\nu(\mathbf{q})|^2 = \xi_\nu(\mathbf{q}) \cdot (\xi_\nu(\mathbf{q}))^*$. The $2 \times 2$ continuum Fourier space dynamical matrix $\mathbf{D}(\mathbf{q})$ has one longitudinal eigenmode along $\hat{\mathbf{q}}$ with eigenvalue $(B + \mu)q^2$ and one transverse eigenmode along $\hat{\mathbf{q}}^\perp$ with eigenvalue $\mu$. We compute $\mathbf{D}(\mathbf{q})_{\alpha\beta} = \frac{1}{N} \sum_j \sum_l e^{iq_\alpha r_{j,\alpha}} H_{jl,\alpha\beta} e^{-iq_\beta r_{j,\beta}}$, where the greek indices $\alpha, \beta$ correspond to $x$ or $y$ and there is no sum implied. We diagonalise the resulting matrix, and choose the longitudinal eigenvector as the one with the larger projection onto $\hat{\mathbf{q}}$, and from there the longitudinal and transverse eigenvalues $\lambda_L(q)$ and $\lambda_T(q)$ after a radial $\mathbf{q}$ average. We fit $B + \mu$ as the slope of $\lambda_L(q)$ vs $q^2$ up to $q = 1.5$, and the same for $\mu$ and $\lambda_T(q)$ (Supplementary Fig. 3).

## Data availability

Data supporting the findings of this manuscript are available from the corresponding authors upon reasonable request.

## Code availability

Simulation and analysis code used in this study are available under an open source (GNU GPL v3.0) licence at: https://github.com/sknepneklab/SAMoS (https://doi.org/10.5281/zenodo.3616475).

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

## Acknowledgements

We acknowledge many helpful discussions with C. Huepe, D. Matoz Fernandez, K. Martens, I. Näthke, R. Sunyer, X. Trepat and C. J. Weijer. S.H. acknowledges support by the UK BBSRC (grant number BB/N009150/1-2). R.S. acknowledges support by the UK BBSRC (grant numbers BB/N009789/1-2). J.M.C. was funded by BBSRC Research Grant BB/J015237/1. K.K. is funded by a BBSRC EASTBIO PhD studentship.

## Author contributions

S.H., E.B. and R.S. developed the theory. K.K. performed the experiment and J.M.C. coordinated it. R.S. developed the SAMoS code used for both particle and SPV model simulations. S.H. and K.K. performed the numerical simulations and analysed the numerical and experimental data. S.H., E.B., R.S. and J.M.C. wrote the manuscript.

## Competing interests

The authors declare no competing interests.

**Additional information**

