## [Peer Review File · Nature Communications]

First round of reviews

Universal motion patterns in confluent monolayers: Response to reviewers

We would like to thank all reviewers for carefully reading our manuscript and providing us with insightful comments and constructive criticism. We address each individual point in detail below; our responses are shown in blue.

Reviewers' comments:

Reviewer #1 (Remarks to the Author):

The manuscript by Henkes et al proposes a theoretical model to study the emergence of collective migratory patterns in cell monolayers. They argue that a confluent cell layer behaves similarly to a jammed or glassy solid, in that the velocity of individual cells can be correlated over long time- and length-scales. They further show that even in the absence of cell-cell alignment interactions, the motion of cells can form swirl and move collectively as if driven by a long-range communication. By fitting data from HCEpC to the theory, they've shown that the model is a plausible mechanism for the observed velocity correlations. Overall, I think it is an interesting piece of work and warrants publication. I would however recommend a careful revision to improve the flow of the text and readability to readers who are not physicists.

Our reply: We thank the reviewer for finding our work interesting and worth to be published, and we address the readability issues below.

I would ask the authors to consider the following comments:

1) The experimental movie taken in HCEpC culture shows there is significant proliferation that occurs over 48 hours. The cell extrusion and divisions appear to be a significant source of active forcing in these cells. Doesn't this result in fluid behavior overall? If so, why is an active solid theory justified here?

Our reply: Our previous work (Matoz-Fernandez, et al., *Soft Matter* **13**, 3205 (2017), ref [58] in the text) indeed shows that any amount of cell extrusion (apoptosis) or division would result in the cell monolayer being in a fluid phase. For dense systems with not too frequent division and extrusion events (as is the case here: the typical cell cycle for human corneal epithelial cells is 60 hours), however, the fluid effects occur over long time scales, exceeding the time scale of persistent motility (approximately 2.2 hours).

The active solid theory is applicable when there is such a separation of time scales between motility and other events leading to rearrangements in the system. We estimate the motility-induced rearrangement time (alpha relaxation time) to be around 3 hours (see Fig. S3), shorter than the division-induced time but of the same order as the motility persistence.

Our experimental system is therefore in the regime where we expect some corrections to our active solid theory due to high motility but not due to divisions, similar to Figure 3b, curve for $T_{eff} = 0.02$ which has $\tau \approx \tau_\alpha$.

Beyond this theoretical argument, we have constructed parameter-matched particle-based simulations of the experimental system (shown in Fig. 4F and 1C) both including and excluding divisions at a rate appropriate to the HCEs. As can be seen in Figures 4F and 4E, the inclusion of divisions at the typical cell cycle rate does not significantly alter any of the observed correlation functions or the mean velocity. We therefore conclude that divisions and extrusions do not play a major role in our system.

We make this point clear in the revised manuscript. Specifically, we have added the sentence “This result is consistent with the observed separation of motility time scales τ and division time scale τ_{div} ”.

2) The metric used to compare theory to experiments should be more accessible to a non-physicist. For example, the correlation functions in Fourier-space can be mystifying to many. How about comparing sizes of swirls, sizes of moving packs and real-space correlation functions?

Our reply: The reviewer raises a good point. We agree that it is not easy to have a good intuitive picture about the meaning of the two-point correlation function in Fourier space. On the other hand, this representation is the most natural way to carry out the theoretical analysis.

We include the real-space correlation function (eqs. 14-15 in the revised manuscript) and the experimental correlations and predictions in Figure 4C, together with labels corresponding to the predicted swirl size in the experiments (Figure 4A).

3) The references [22, 37, 48] have also reported similar patterns of collective migration in various active material systems. Especially in reference [22] Henkes and co-workers used a very similar approach to understand why migratory swirls look like low frequency excitations in a solid. A more detailed discussion on what sets the current work apart is needed.

Our reply: The key difference and novelty of this work compared to the previous studies mentioned by the reviewer is that this is the first time the formalism of dense assemblies of active particles is applied to cell monolayers.

Ref. [22], in particular, is a theoretical/numerical study of a model system of active self-propelled disks with no reference to experiments. The alignment mechanism treated in Ref. [22] assumes a level of (mechanical) planar cell polarisation where the preferred migration direction of the cell reacts to its mechanical environment. Using the open sheet environment we are in, it would predict collective flocking migration of the cells, which we do not observe.

For the revision, and also in response to referee 4, we considered sub-threshold amounts of the alignment mechanism of Ref. [22] to understand if the faster drop in the velocity correlation function of the experiment compared to the predictions can be explained this way (see Fig. 2 below). We found this not to be the case, as a substantial drop could only be achieved in a regime of full sheet migration (which, again, we do not observe).

References [37, 48] indeed make similar observations but do not provide a theoretical description of the mechanism behind the observed collective motion patterns. Here, on the other hand, we provide theoretical arguments for the observed swirl patterns and validate them by a direct comparison to experiments and numerical simulations on particle-based and Voronoi-based numerical models.

4) The abstract claims the model can fit the experimental data without any parameters. This does not appear to be the case as the shear and bulk moduli, τ , v_0 are all fitting parameters.

Our reply: The prefactor $(v_0/\bar{v})^2$ in eq. 12 (eq. 16 in the revised manuscript), as shown in eq. 11, depends only on the correlation lengths of velocity fluctuations and a microscopic cutoff, assumed to be of the order of the average cell size. Furthermore, if one makes a plausible assumption that the ratio $(\mu + B)/\mu$ is the same as in simulations of soft disks, only one free parameter remains. As noted in Sec. IV, we choose the transversal correlation length of velocity fluctuations as the single fitting parameter. We note that individual values for elastic constant B and μ cannot be reliably obtained from our experiments.

We revised the abstract and Section IV to clarify this point, and we correct the unintentionally misleading language in the original abstract: it is the simulated data that we can fit without parameters.

5) As the authors state, there are other proposed interaction forces between cells including flocking, nematic and contact-inhibition of locomotion that can also lead to similar behavior. Are there ways that would distinguish the patterns caused by mechanisms proposed here versus those caused by e.g. flocking? Rather than only showing the experiment is consistent with the model, it would be helpful and much more impactful to suggest experimental tests that would eliminate other possible mechanisms.

Our reply: The reviewer raises a very important point:

1. Regarding a flocking interaction, as explained above, if of sufficient strength, this would lead to collective sheet migration, which we do not observe. We believe that there are at best very small amounts of flocking alignment in our system, but we are not able to exclude its presence entirely (see Fig. 1 in the answer to reviewer 4 below). According to the phase diagram of Giavazzi, et al. (Soft matter **14**, 3471 (2018)), our system is either in the disordered region, or flocking is entirely absent.

To directly answer the question: If flocking were prominent, the sheet would migrate collectively.

2. Regarding contact inhibition of locomotion: The result of such a mechanism is a marked slowdown of individual crawling motility at high density. Arguably, this is exactly what our model predicts – the mean square cell velocity drops precipitously at significant persistence times. In our model, this is due to elastic deformation of particles/vertex polygons. It's worth remembering at this point that particle models and vertex models are both effective theories – the energy functions associated with them should always be thought of as arising from the natural active behaviour of the cells together with the passive mechanics of the cytoskeleton. While our model does not include an explicit mechanism for contact inhibition of locomotion, its simple formulation is nevertheless compatible with it.
3. The recent results pointing to a nematic component in epithelial cell migration on a substrate square with our result as follows: We believe that any nematic behaviour in our system is in addition to our disordered motility, and subdominant compared to it. In the experiments of [4], the nematic behaviour (defect motion and cell extrusion at $+1/2$

defects) is a statistical result. The mean cell motion is dominated by fluctuations. Our result precisely describes these fluctuations due to the disordered part of cell motility. Unfortunately, the limited field of view and image quality of our experiments did not allow us to perform the nematic analysis of Doostmohammadi et al., and we would welcome more experiments! It will be important to disentangle the role of different mechanisms of activity (crawling and apical contraction) in the cell sheet.

The important open question that needs to be answered in the future is how disordered motility (studied here), nematic activity and flocking interact to give rise to the observed patterns of collective cell motion.

Experimental tests that would distinguish between different interactions forces are not easy. We speculate that a careful study that would interfere with signalling pathways responsible for mechanotransduction or expression of genes that control various aspects of cytoskeleton and/or cell-cell adhesion would provide valuable insights. This is, however, beyond the scope of this work.

6) There are a few places that mention cell-level heterogeneity, what exactly is meant by this? Please clarify.

Our reply: This is just rather loose wording for stating that individual cells are not identical to each other, i.e., do not have exactly the same mechanical properties (a commonly used assumption in model simulations of active matter systems). We introduce heterogeneity into our models by including 30% of polydispersity in the cell radii. Since cells are at different points in their individual cell cycles, this is a reasonable assumption. Stochastic differences in gene expression may also be responsible for further differences between cells. In the revised manuscript we make this point clear.

7) It appears that the effective temperature T_{eff} is only well defined for small values of τ , i.e. when the rotational noise can be considered as a white noise. It is a well-defined quantity when τ is large, where the system must have broken fluctuation-dissipation relations. I would suggest just use v_0 and τ .

Our reply: We agree that T_{eff} is not a good “temperature” for large values of τ as fluctuation-dissipation is definitely broken: As eq. 5 shows, the equipartition is no longer satisfied. However, using simply v_0 and τ places the glass transition mainly on a diagonal of constant T_{eff} , making it difficult to compare equivalent simulations to each other. In other words, the relatively slow shift of the glass transition to higher T_{eff} at moderate τ makes this still the most convenient parameter, in spite of its lack of genuine thermodynamic interpretation. As was recently shown by Gov et al. (ref. [46] in the text), RFOT models predict to first order a linear shift in the active glass transition with τ at constant T_{eff} . Our results are qualitatively consistent with this, and we now briefly discuss this point in the manuscript.

8) How is the dynamical matrix in the active vertex model calculated?

Our reply: We do not compute the dynamical matrix for the active vertex model: Figure 3C only includes the numerical data and the continuum predictions, with estimates for the moduli taken from the study of Sussman et al. (ref [56] in the text).

9) μ is used as an index for eigenmodes as well as shear modulus. This should be changed to avoid confusion.

Our reply: Thank you for pointing this out. Following the reviewer's comment, we have changed the index of the eigenmodes to ν .

10) It is known that the shape of the confinement boundary can influence the migratory pattern of cells. How does the artificially added boundary tension in the vertex model influence your results? If at high τ values the system tends to funnel energy into the lowest modes, then do you observe globally rotating cell colonies?

Our reply: We believe that our boundary tension (which is also effectively viscoelastic since new nodes are continuously added or removed) is sufficiently small to not significantly influence migration patterns, and we note that real cell colonies include a boundary actomyosin cable which has a similar effect. In fact, at low boundary tensions we see a fingering instability similar to that observed in real cell colonies, and we excluded that effect on purpose. Note that our correlation lengths only reached system size at the largest value of $\tau = 2000$. We therefore do not observe globally rotating cell colonies.

11) Labels for the y-axis in figures S2 and S3 need to be corrected.

Our reply: Thank you for pointing this out. We have fixed the labels.

12) The subtle difference in the definitions of Eq 6, 10 and 11 should be spelled out to avoid confusion.

Our reply: In the revised manuscript we make these differences clear.

13) In the second sentence following Equation (5), "In the opposite, high activity limit". Is this correct? Do you mean "high τ limit"?

Our reply: Indeed, large τ corresponds to large activity. This is due to τ controlling the time scale over which the motion of a particle is persistent (no turns). We make this point clear in the revised manuscript.

Reviewer #2 (Remarks to the Author):

Henkes et al present an analysis of velocity fluctuations in a linear elastic model with friction and active, time-correlated noise. They calculate the velocity correlation function for such a system and find that it involves a length scale that depends on the elastic moduli of the system, the friction coefficient and the time scale of noise correlation. The structure of this correlation function at equal time is observed in simulations of a soft-disk model and active vertex model; when the viscous relaxation time of these models is large enough. The authors analyze experiments of a fluctuating biological epithelium and find a similar structure to the correlation function at equal time.

Overall I find that the analysis is very seriously done and the paper makes an interesting

point. My main comment is that the wording could be clarified and made closer to the actual results which are more straightforward than the text makes it sound. In the language of coarse-grained theories, the authors find a microscopic correlation time scale is generating a microscopic correlation length scale, which is not surprising. What I find interesting is the point that this length scale is experimentally visible when the viscous relaxation time of the system is sufficiently large.

Our reply: We thank the reviewer for a positive assessment of our work.

Other comments:

1) The word “universal” seems exaggerated. A number of very strong assumptions are made about the system that the theory describes.

Our reply: We agree with the reviewer that the ‘universal’ language can be confusing as it has a very specific usage in the statistical mechanics community. We have rephrased to abstract and changed the title to “Dense active matter model of motion patterns in confluent cell monolayers”.

2) The correlations are not “long-range” since they decay exponentially.

Our reply: We appreciate the point the reviewer is raising. In the strict statistical mechanics sense, only power-law correlations are considered long-range. Here, however, we use the term more loosely and refer to long-range correlations as those for which the correlation length significantly exceeds the cell size, and can even reach the order of the system size. In the revised manuscript we make this point clear.

3) It seems to me that the length-scale discussed in the manuscript is not “emerging”; in fact it is the opposite; it does not emerge on large length scales but arises from a microscopic characteristic time.

Our reply: This is again, to some extent, a problem of phrasing. Indeed the length scale depends on a microscopic persistence time that is large, but the link between the large persistence time of a single cell and the correlation scale is through the collective, elastic dynamics at the many-cell level. In this sense, we consider the length scale as emerging. A single cell moving on a substrate would still have a large persistence time τ , but the only length scale that can be associated to it is the persistence length $l = v_0\tau$, which is linear in the persistence time τ , while the collective correlation length scales as the square root of τ .

However, we agree with the referee that it is more precise to say that the correlated microscopic rotational noise together with soft elastic cell-cell repulsions leads to the onset of correlated motion pattern at scales spanning multiple cell sizes. We have rephrased the text to make this point clear.

4) Eq. 9 does not follow directly from Eq. 40 in the supplementary as there is a Dirac delta of $q+q'$ which should evaluate to infinity for $q'=-q$. Similarly the relation between Eq. 11 and Eq. 49 in the supplementary has the same problem. Can the authors clarify their notation?

Our reply: We thank the reviewer for pointing out this issue. It is related to the two definitions used for the Fourier transform, i.e., a discrete Fourier series for finite systems and continuum

Fourier transform for infinite systems. In the continuum mechanics analytical calculations, it is more convenient to use a continuous Fourier transform. However, when comparing to simulations of finite-size systems, it is natural to use the discrete Fourier transform. There are some subtle technical differences, such as those pointed by the reviewer, that are carefully discussed in the Supplementary Information (eqs. (29) to (32) and discussion around those equations). Being able to switch from one notation to the other is indeed essential to compare analytical and numerical results. The precise correspondence between the two definitions is given in eq. (31) in the SI.

In the revised version, we kept eq. 9 as it was, but we have clarified how it is obtained from eq. 40 of SI [now eq. 41] by explicitly writing the transformation for the discrete to the continuous Fourier transform, see eqs. 42 and 43 of SI.

5) Before section 3 I would suggest to give the expression of $G(q)$

Our reply: We have replaced the notation $G(q)$ by the more explicit notation $\langle |v(q)|^2 \rangle$, to avoid confusion.

Reviewer #3 (Remarks to the Author):

I have fairly mixed feelings about this paper, 'Universal motion patterns in confluent cell monolayers', by Henkes et al. The main contribution of this work is a derivation of the (fourier space) velocity correlation function of any overdamped system of active units near a phase space point of local mechanical equilibrium. In this context, and for a particularly simple form of out-of-equilibrium dynamics, they are able to connect the quantities of interest to an expansion of the normal modes of the system. On its own, I think this is an interesting contribution, although perhaps not one that on its own justifies publication in Nat. Commun. (in part since it is a natural extension of previous work by some of the same authors -- see cited references in the main text).

Our reply: While this work builds on our previous results, we think that it is a significant step forward and not just an iteration of an existing story. Specifically, to the best of our knowledge, this is the first study of this kind that provides a physical mechanism for the observed swirl-like motion patterns in cell monolayers. Our results are robust and account well for both experiments and two very different types of simulations (particle-based and Voronoi-based). Our results are universal in the sense that the proposed theory does not depend on the details of the specific cell type but is a generic feature of all confluent epithelial sheets.

We, however, agree that the word "universal" has a broader meaning than we intended here and therefore we changed the title to "Dense active matter model of motion patterns in confluent cell monolayers".

The paper then makes many statements whose validity I am unsure of, beginning with the very title of the work. What exactly is 'universal' in what the authors have done? The simulations the authors have done are of soft disks and 'vertex models', and as far as I can tell from reading the literature these models apparently have both a different normal mode spectrum & a different type of 'glassy' behavior. How universal is the functional form of the

velocity autocorrelation function when different models of non-equilibrium dynamics are considered?

Our reply: Our theoretical model only assumes the *existence* of a normal mode spectrum, and for the analytical prediction, that at low q it scales as predicted by the theory of elasticity. Both assumptions are perfectly justified for both the soft disks and the vertex models, as long as the system remains away from the glass / jamming transition region. To be slightly more technical, what allows us to get away with this is the strong amplification of the low q region for persistent driving: The high- q region, where the normal mode spectrum is of course very different for soft disks and for vertex models, does not contribute significantly.

When we choose packing fractions (soft disks) or shape parameters (vertex models) that are very close to the glass transition, the normal mode spectrum loses its low q standard scaling due to the appearance of a boson-like peak (or excess of low frequency modes). We purposefully stay away from this regime.

We note that actual cell sheets are confluent, i.e. densely packed, and appear very far from a density/shape driven glass transition. We do see fluidisation, of course, but this is due to the active driving, which as we established is quite unlike a thermal glass transition.

What is striking is that the correlation functions we computed in this work depend only on the active dynamics and not on the details of the steric interactions as supported by the fact that simple particle-based simulations have the same autocorrelation functions as a far more complex Voronoi (Vertex)-based model (with the same active dynamics). We argue this to be the key strength of our findings.

Regarding the type of non-equilibrium driving: This is an intriguing question that is worth exploring but it is beyond the scope of this work. The advantage of the proposed active dynamics, i.e., uncorrelated random noise of the cells' directions, is that it is the only one that can be studied analytically in a relatively straightforward way. Even the simplest extension where each cell aligns its direction to its instantaneous velocity requires some strong simplifying assumptions to allow for analytical treatment. More involved models would only be treatable numerically.

Or when the equations of motion are not overdamped? Or...

Our reply: All experimental systems that are relevant to this work are in the very low Reynolds number regime. Exploring the regime where the inertial effects play a role is interesting for, e.g., dense crowds, but would not be relevant to describe cell-size experimental systems, as we aim to do here.

The abstract claims that uncorrelated, persistent motility that couples to the collective elastic modes of the cell sheet are sufficient to produce the characteristic 'swirl-like' patterns seen in the experiments. Are the authors not concerned about Fig. 4A, which seems to show the biggest deviation between theory and experiment is for wavevectors that correspond to 1-10 cell radii -- *precisely* the scale of the 'swirl-like' patterns?

Our reply: Figure 4A indeed shows some discrepancy between the model and the experiment. Note however that it does show that above a length scale of 5-6 cell radii – the swirl

size – theory and experiments agree with each other. This is all that our mesoscale elastic model ever claimed.

As to the origin of the discrepancy: We have so far excluded experimental imaging problems, local alignment between cells (see Figure 2 below) and finite size effects. We believe the issue is likely due to neither disks nor Voronoi cells being a particularly good model for the very local dynamics of (these) cells, due to, e.g., the absence of tangential friction terms. This is something for the community to look into!

To further show that our model is actually a good fit for the data, we have added the real-space correlation functions and our new analytical predictions for it (eq. 14) in Figure 4C. Using the same fitting parameters as in Fig. 4B (Fourier space), it becomes clear that at scales above a couple of cell radii (11 microns), the model fits.

Some relatively more minor concerns:

The abstract claims that the experimental data is fit without any free parameters. Perhaps this is a semantic distinction of what exactly constitutes a ‘free’ parameter, but after Eq. 12 the authors seem to be (a) artificially (?) fixing the ratio of elastic moduli they *assume* the experimental sheet to have, and then (b) fitting a value for ξ_T^2 to achieve a curve closest to the data. This seems like a free parameter.

Our reply: As discussed in our responses to the reviewers 1 and 2, wording in the abstract was not sufficiently accurate and we revised it. We had intended to say that we fit the numerical data without free parameters.

I find some of the writing confusing. To take a few examples... the abstract has a sentence that starts ‘This includes a divergent correlation length...’. The sentence is manifestly about the theory, but it immediately (and conceptually) follows a sentence about matching with experiments.

Our reply: We made this point clear in the revised abstract.

Similarly, the authors initially say they are simulating vertex models, but then after Eq. 1 it turns out that they require F_i^{int} to be expressible as the gradient of an energy function that depends only on cell centers (which the vertex models of Honda et al. are not). It turns out they are simulating some variant of the modern ‘Voronoi’ models, but have adopted a confusing nomenclature.

Our reply: We indeed simulate a self-propelled Voronoi model (also referred to as the Active Vertex Model, ref [38]), as stated in the Methods section. We now refer to the SPV (self-propelled Voronoi) model throughout when referring to our vertex model.

In section III they chose some particular value for the dimensionless shape factor to ‘put the passive system into the solid part of the phase diagram’. Two questions: (1) How sensitive are their results to this parameter choice?

Our reply: The results are quantitatively sensitive to this parameter choice (through the values of the shear and bulk modulus), but not qualitatively, as long as we stay below the glass / jamming transition region $p_0 \approx 3.81$. For example, we have repeated the experimental fit with

$p_0 = 3.8$ with no major qualitative changes except a shorter alpha-relaxation time in the simulation.

(2) When I read their quoted references [37,54], it seems that \emph{any} choice of dimensionless shape factor would put the no-activity limit of the system into a weakly solid phase. This seems strange to me, but this is also the reference that suggests that the normal mode structure of the Voronoi model might be very different from the soft spheres... what is going on?

Our reply: That is an excellent question, and we do not think that the behaviour of the self-propelled Voronoi model for $p_0 > 3.81$ is properly understood.

In this region, $p_0 > 3.81$, it becomes very unclear if the self-propelled Voronoi model is appropriate for cells. For example, it creates both quasi-rosette structures (ref. [38]), and junctions with no tension on them which should then be fluctuating instead of straight [L Yan, D Bi, Phys. Rev. X 9,011029 (2019)]. As the referee points out, it was also recently shown that the normal mode spectrum in this region is anomalous (ref. [54]). We did not study this region as we do not believe that it can safely be considered as an appropriate cell model.

Reviewer #4 (Remarks to the Author):

In this manuscript, the authors tackle the question of swirling patterns in cultured cellular layers, a topical problem which has received considerable interest in the past few years. They perform a mixture of analytics/active matter theory with several types of numerical simulations of tissues, to explore the implications of purely uncoordinated active motions on large-scale movements. They then verify some of the predicted features on experimental data. Overall, the manuscript is well-written, and the approach is interesting for a wide community, aiming at exploring a very simple explanation of correlations in confluent monolayers.

I do have a number of comments/concerns however on the way that the authors compare theory and experiments, as well as the strength of some claims, that would need to be addressed.

Our reply: We are happy that the reviewer finds our work worth publishing, and we would especially like to thank him/her for the constructive criticism below, which has led to substantial improvements in the paper.

Major points:

- I would tone down some of the claims of the article. The mention of “universal” in the title, associated to an experimental characterisation, creates an expectation that the authors will look at different cell types, different densities or conditions. In fact, it refers to features that are common to several numerical models considered (+analytics), which is very interesting, but different from the impression the current title gives. Similarly, I found the claim of fitting without any free parameters from the abstract a bit too strong: the authors do have to fit several parameters from the data before explaining the velocity functions.

Our reply: We agree with the reviewer and with the other three reviewers that language in the abstract is at places too strong. In the new version of the manuscript we have completely

revised the abstract, and our new title is now “Dense active matter model of motion patterns in confluent cell monolayers”.

On that note: as mentioned by the authors, single cells in that model would perform a persistent random walk, close to what is observed. Are the fitted parameters consistent with single cell behaviour? Could that be a way to constrain the model much further? (leaving only collective elastic interactions as unknown)

Our reply: We agree with the reviewer that this is a good idea, and we have attempted to measure just this by plating individual HCEs on fibronectin and then measuring their dynamics. Unfortunately, we do not have a cell line with a fluorescent nuclear marker (several attempts to transfect the HCEs failed), and extracting reliable traces from simple phase contrast images proved difficult. Figure 1 (left) shows representative tracks from over 10 hours imaging of cells at approximately 10-20% confluence, using the trackpy analysis program. While we can extract MSDs (right), even after removing short traces and non-sticking cells, the values range over two orders of magnitude, and are well below the expected persistent random walk curve for $v_0 = 90 \frac{\mu m}{h}$, $\tau = 2.2h$. While some of this is appropriate for the small cell colonies we tend to observe, we do not think that these experiments prove anything either way. Instead, we note that Garcia et al. (ref. [40]) have measured low-density speeds in epithelial cell colonies, using PIV, and find speeds consistent with our predicted value of $v_0 = 90 \frac{\mu m}{h}$.

In addition, this procedure is strictly only appropriate for physical active particles, which interact with each other simply through repulsive forces that have no effect on the self-propulsion mechanism. But biological cells can be modelled as such active particles only in a specific context, as interactions between cells also affects their internal dynamics, e.g., the appearance/disappearance of lamellipodia. So looking at the dynamics of an isolated cell does not necessarily provide relevant information on the behaviour of cells in the dense regime.

Figure 1: Tracking low density HCEs on fibronectin. Left: Cell tracks from trackpy, showing difficulties associated to imaging without fluorescence. Right: Individual MSD curves extracted from these tracks.

- Maybe a more accurate claim for Fig. 4A is that the theory gives the correct scaling form (which is already very interesting). Is it still the case though for Fig. 4D? I realise that this is a numerics vs. exp type of fitting, so the question is tougher, but it is slightly confusing in the paper that the authors introduce $P(v)$ there in the text without having created a theoretical/numerical intuition for it beforehand (as they did well for $v(q)$)... I think it would help the paper if the authors plotted $P(v)$ in Fig. 3 for the same range of parameters they do for $v(q)$. This would help understanding what is parameter-dependent in this

plot, and what is generic (for instance the slopes on both sides of the peak). This would serve as a platform for an extensive discussion on the topic in main text (right now, the authors only briefly mention potential explanations for deviations at large v for instance).

Our reply: We agree with the reviewer that a more detailed understanding of the measured distribution $P(v)$ would be useful. However, we were not able to give simple theoretical arguments to account for the main features of the distribution, so we kept the discussion of this distribution to a minimal level. Overall, the emphasis of our study is really on the spatial correlations of the velocity field, and the distribution of velocities is rather considered as a side result at this stage.

- I realise this might be out-of-scope for this paper, and therefore not strictly necessary, but i think the manuscript would be enriched if the authors considered quantitatively an alternative to their model. For instance, if they took active alignment with very low persistence time, would they still be able to fit the data, or is the dependency of Eq. 12 really exclusively specific to uncoordinated motion? Proving that alternative models wouldn't fit as well would make the claim even stronger (although again, i recognise the strength in showing that simple models explain the data).

Our reply: A similar issue was raised by reviewer 1, above. In short, we do not observe any flocking in the experiment, so that alignment interaction has to be weak if it is present. We checked that including a weak alignment with a low persistence time does not significantly change the results with compared having no alignment at all. Our model is the same as in ref. [22]:

$$\dot{\theta} = \frac{1}{\tau_v} \sin(\theta - \theta_v) + \eta, \quad \langle \eta(t)\eta(t') \rangle = \frac{1}{\tau} \delta(t - t'),$$

where θ_v is the angle of the velocity vector. Here there are two angular time scales, the noise time scale τ (2.2 hours for our experiments) and the alignment time scale τ_v , which compete with each other.

As shown in Figure 2 below, when $\tau \ll \tau_v$, the results are virtually indistinguishable from the case with no alignment. Only when the time scales are of similar magnitude do we see changes in the correlation functions. However, as the order parameter plot shows, this coincides with the appearance of global flocking in the system, which we do not observe.

Figure 2: Effect of velocity alignment on the experimental fits. Top left: Fourier space correlation functions, the 'no_align' curve is the one from the paper. Top right: Order parameter plot, showing alignment only for $\tau_v \leq \tau$ (note that the minimum order parameter values show finite size effects). Bottom right: Velocity autocorrelation functions, again showing significant effects only for $\tau_v \leq \tau$.

- the experimental variability in velocity autocorrelation functions is relatively surprising. For instance, exp 5 seems to be correlated with extremely fat tails (Fig. S1B), although its characteristic velocity looks reasonable, as well as its other metrics (Fig. 4)? Could the authors comment on that, and possible confounding factors? For instance, they mention an average measured cell density in the main text, but don't comment on the dispersion. Could it be that exp 5 is at much higher density? I'm also asking because according to their own prediction, a much higher value of τ for a given experiment should manifest in the other predictions shouldn't it?

Our reply: We'd like to thank the reviewer for pointing this out. Unfortunately, we do not have good automated cell counting for our system (several attempts to develop a permanent fluorescent cell nucleus marker for our HCEs have failed). Manual counting did not show significant differences, and snapshots do not look significantly different (see Figure 3).

Figure 3. Representative snapshots from the experiment. Left: Experiment 2, behaving typically. Right: Experiment 5, with longer correlation times.

We have however noted the longer autocorrelation times in experiment 5, and also in experiment 6 (which has now been included in the paper). These longer correlation times also correspond to longer-ranged spatial correlations (see the new Figure 4C), and we can get reasonable agreement with the theory using the same elastic moduli and a longer τ . However, as Figure 1 shows here, alignment can also lead to longer autocorrelation times, and we cannot exclude that some is present in these two experiments; on the actual cornea PCP does lead to long range migration, after all.

- The introduction of the paper reviews well the literature on this active topic, however, the bulk of the text could be better discussed in its context. For instance, Nandi et al, PNAS, 2018 have examined the role of persistence time on glassiness in models of persistent active particles without alignments (also finding subtle effects). Garcia et al, PNAS 2015, have also reported explicitly that correlation lengths in velocities arise as a result of activity, even in the absence of alignment forces.

Our reply: We thank the reviewer for pointing out these two papers!

Nandi et al. (2018), and indeed the recent Mandal et al. (2019) both discuss the active glass transition in the limit of high active persistence time. Our results in Fig. 2 are qualitatively consistent with the results of the RFOT theory, in that the glass transition border roughly scales as $T_0(1 + c \tau)$, i.e., the passive glass transition temperature shifts upwards with persistence time. We now cite both papers and discuss the connection to these results in section III.

We would also like to thank the reviewer for making us take a closer look at Garcia et al. (2015). In addition to detailed experimental results, this article introduces a scaling theory for the correlation length and the mean velocity for cells dissipating either primarily with a substrate or each other. Our results exactly match the results of Garcia et al. in the region dominated by cell-substrate interactions: Our theory, which is for purely cell-substrate dissipation, reproduces the $\langle v^2 \rangle \sim 1/\xi^2$ scaling, with additional log corrections (eq. 11 in the manuscript and Fig 3d). For our simulations, we find a driving speed of $v_0 = 90 \mu\text{m}/h$ that then reduces to $\sim 10 \mu\text{m}/h$ in the dense phase. This is identical to the low-density and high-density data reported by Garcia et al. The peak correlation length, $\xi = 100 \mu\text{m}$, is also the same, and so are our predicted exponential real-space correlation functions.

- in the experimental movies, bright extruded cells seems to be accumulating in time, meaning that the initial picture looks very different from the first one. Can the authors show that averaging in time is still ok in this case? (i.e. that the relevant quantities are not ageing/time-dependent in experimental settings?)

Our reply: We were concerned by the same issue, and prior to our first submission we performed the following check: We divided each experiment into two halves, early and late. After excluding the first two hours of footage (cells getting used to the new environment of the microscope), the analysis performed on the two halves of each run gives indistinguishable results. Note that our cells come pre-aged: The wells already reached confluence days to weeks beforehand and were then stored.

Minor points:

- Why are experiments skipping 4 and 6? The authors mention in methods that exp 4 was not used in the paper, was there a reason for the exclusion?

Our reply: We skipped experiment 4 because it did not follow the same protocol: Instead of using one of our stored confluent cell layers, the cells from experiment 3 were imaged for a further 48 hours. This resulted in (eventually) a noticeable slow-down of cell speed, something we do not observe in our other experiments.

Experiment 6 was somewhat of an outlier, in that it showed stronger variations in the mean cell speed over time, and we were concerned some external disturbance, e.g. in the microscope conditions. Its results are nevertheless consistent with the other data presented here, and we have now added experiment 6 back into the paper. Its properties are intermediate between experiment 5 and the “typical” experiments 1,2,3 and 7.

- At first read, i got confused by the absence of peaks in Fig. 4A, before realising that the authors were cutting the x-axis as they want to concentrate on the small q regime, which is logical given the scope, but am not sure if it's explicitly mentioned? It might be interesting to show the full functions experimentally in Supplementary? I wondered in particular if comparing them to the numerics, which do capture this structure, might be helpful in some ways? (in particular viz. the small discrepancy at larger q for Fig. 4A).

Our reply: We have not cut off the large q regime in our experimental correlation functions, there are simply no peaks present! We have also used a finer-meshed PIV grid to investigate the issue, and not seen an upturn in the correlation functions. As can also be seen in Figure 4F, our simulated fits with the correct radii do show peaks at a q corresponding to the inverse cell size. We therefore think that either the imaging becomes unreliable at these scales (possible but unlikely, this is well above the microscope's resolution), or that the model becomes unreliable at these scales. The likely culprit is the model, in particular the amount of tangential slipping motion at cell scale in both models which pair-friction in a real experimental system would limit. As already mentioned in our response to reviewer 3 above, this is a challenge for the field, and we now mention this issue in our conclusions.

Second round of reviews

Reviewer #1 (Remarks to the Author):

The revised manuscript by Henkes and coworkers have improved in terms of readability and accessibility to non-physics readers. The authors have also addressed my technical concerns. I therefore recommend it for publication.

One additional comment regarding the title change. My opinion is that the authors should not shy away from claiming this to be a universal phenomenon. Even though there are a few works (as referenced in the introduction) that have suggested possible mechanisms for swirl-like motion in active matter (without explicit flocking-type interactions), this paper offers a concrete theoretical foundation for the observed collective *universal* behavior. I disagree with the other two reviewers that word 'universal' and assumptions taken in the theory are too strong. The remarkable thing here is that the theory literally only assumes that cell layers should support elastic modes.

Our reply: We thank the reviewer for recommending our manuscript for publication. After discussing it with the editor we concluded that it would be preferable not to use the word "universal" in the title.

Reviewer #2 (Remarks to the Author):

The authors have answered adequately to my comments, I recommend publication of the manuscript.

Our reply: We thank the reviewer for recommending our manuscript for publication.

Reviewer #3 (Remarks to the Author):

I'm afraid my very mixed feelings about this paper remain unresolved. First, I would like to thank the authors for the very serious work and consideration with which they have tried to address both my own criticisms and the comments of the other referees. Second – and I realize that with three otherwise largely positive reviews I am unlikely to be persuasive – I continue to wonder the extent to which the claims in the paper are justified. I continue to be confused, also, by whether the authors are making a claim just about the long-length-scale elastic behavior, or about anything non-trivial in the normal-mode structure that leads to the types of meso-scale displacements observed in the experiments. Particularly given that, in the parameter regime that seems to fit best, the "activity" is basically indistinguishable from an equilibrium system at an effective temperature, I agree with the first referee that a more detailed discussion of what sets this work apart would be needed, as certainly other researchers have proposed thermal projections onto vibrational modes as an origin of the dynamic heterogeneities seen in cellular monolayers.

Our reply: We strongly disagree regarding the referee's claim that our system behaves as a thermal, equilibrium system. This is directly seen from the systematic breaking of energy equipartition between normal modes. For large persistence time, the average elastic energy

stored in a mode is proportional to λ^{-1} , where λ is the stiffness of the mode, which is very different between soft and stiff modes. In contrast, effective energy equipartition and thus equilibrium-like behaviour is only observed in our system for very small values of the persistence time. In the soft disk model parametrized from the experimental data, we obtain a persistence time $\tau = 2.5h$, and a passive relaxation time of the overdamped elastic dynamics $\left(\frac{k}{\zeta}\right)^{-1} = 0.02h$, approximately. Hence the persistence time is a hundred times larger than the passive relaxation time, meaning that we are in the large persistence time regime, that is, far from thermal equilibrium in any reasonable sense. Assuming thermal statistics for the normal modes would lead to a strong discrepancy between theoretical and numerical/experimental velocity correlation functions.

The confusion might arise from the fact that we use an effective temperature parameter to describe the glassy phase. As explained in our previous reply, we do so because this effective temperature is a convenient parameter to describe the phase diagram of the glass transition. However, we emphasize again that this effective temperature is by no means an effective thermodynamic temperature, it is just a convenient parameter having the dimension of a temperature.

On the other hand, we agree with the referee that the claim of generic behaviour amongst different systems regards the long-length-scale behaviour, that can be described within the framework of continuum elastic theory. This is in itself a new and valuable tool to investigate such systems.

However, a more accurate description (down to cell sizes) of a given system in the low motility and large persistence time regime is given by the normal modes expansion, retaining the full spectrum of modes, and which is in complete agreement with our simulated results.

Reviewer: A few short individual responses follow:

I have fairly mixed feelings about this paper, “Universal motion patterns in confluent cell monolayers”, by Henkes et al. The main contribution of this work is a derivation of the (fourier space) velocity correlation function of any overdamped system of active units near a phase space point of local mechanical equilibrium. In this context, and for a particularly simple form of out-of-equilibrium dynamics, they are able to connect the quantities of interest to an expansion of the normal modes of the system. On its own, I think this is an interesting contribution, although perhaps not one that on its own justifies publication in Nat. Commun. (in part since it is a natural extension of previous work by some of the same authors – see cited references in the main text).

Our reply in the first round: While this work builds on our previous results, we think that it is a significant step forward and not just an iteration of an existing story. Specifically, to the best of our knowledge, this is the first study of this kind that provides a physical mechanism for the observed swirl-like motion patterns in cell monolayers. Our results are robust and account well for both experiments and two very different types of simulations (particle-based and Voronoi-based). Our results are universal in the sense that the proposed theory does not depend on the details of the specific cell type but is a generic feature of all confluent epithelial sheets. We, however, agree that the word “universal” has a broader meaning than we

intended here and therefore we changed the title to “Dense active matter model of motion patterns in confluent cell monolayers”.

Reviewer’s response: The statement that this is the first study that provides a physical mechanism seems deeply unfair to a large body of literature in the field. Some of the early papers quantifying dynamical heterogeneity in cell sheets tried to quantify properties of an effective dynamical matrix and found peaks in both frequency and wavevector space. The idea that low frequency and quasi-localized modes of the dynamical matrix contribute to either rearrangements or large-amplitude displacements at low temperature and form a good basis for projecting motion has been well-appreciated in the jamming community for some time, and the fact that, indeed, most of the data has been presented in a regime where the large-persistence time, very non-equilibrium aspect of the calculations seem not to matter (i.e., estimating $\tau \approx 2.4h$ in units of time given by $\zeta(K\langle A \rangle)^{-1}$ to be pretty small) suggests that low-T projections onto normal modes is message of this paper; if so, the novelty of the work is not so great in this regard.

Our reply: We again have to respectfully disagree with the referee. While there is indeed a number of studies that have quantified dynamical heterogeneities in cell sheets, to the best of our knowledge, no study to date has made a connection between cell-level self-propelled behaviour and collective cell migration patterns in confluent cell monolayers. We do not disagree that some of the physics discussed here is intimately known to the jamming community. It has, however, not been applied to cell monolayers, which, unlike the passive colloidal suspensions usually studied by the jamming community, are a prime example of a dense active system.

As we pointed out above, by the argument of the referee, we are actually in the strongly active regime, i.e. $\frac{\zeta}{K\langle A \rangle} = \left(\frac{k}{\zeta}\right)^{-1} \approx 0.02h \ll 2.5h$. This is also consistent with our position on the dimensionless plot Figure 3d at $\frac{\xi T}{a} \approx 5$, $\frac{\langle v \rangle}{v_0} \approx 0.1$, on the right, strongly active side.

Reviewer: The paper then makes many statements whose validity I am unsure of, beginning with the very title of the work. What exactly is “universal” in what the authors have done? The simulations the authors have done are of soft disks and “vertex models”, and as far as I can tell from reading the literature these models apparently have both a different normal mode spectrum and a different type of “glassy” behavior. How universal is the functional form of the velocity autocorrelation function when different models of non-equilibrium dynamics are considered? Or when the equations of motion are not overdamped? Or...

Our reply in the first round: Our theoretical model only assumes the existence of a normal mode spectrum, and for the analytical prediction, that at low q it scales as predicted by the theory of elasticity. Both assumptions are perfectly justified for both the soft disks and the vertex models, as long as the system remains away from the glass/jamming transition region. To be slightly more technical, what allows us to get away with this is the strong amplification of the low q region for persistent driving: The high-q region, where the normal mode spectrum is of course very different for soft disks and for vertex models, does not contribute significantly. When we choose packing fractions (soft disks) or shape parameters (vertex models) that are very close to the glass transition, the normal mode spectrum loses its low q standard scaling due to the appearance of a boson-like peak (or excess of low frequency modes). We purposefully stay away from this regime. We note that actual cell sheets are

confluent, i.e. densely packed, and appear very far from a density/shape driven glass transition. We do see fluidisation, of course, but this is due to the active driving, which as we established is quite unlike a thermal glass transition. What is striking is that the correlation functions we computed in this work depend only on the active dynamics and not on the details of the steric interactions as supported by the fact that simple particle-based simulations have the same autocorrelation functions as a far more complex Voronoi (Vertex)-based model (with the same active dynamics). We argue this to be the key strength of our findings.

Reviewer: It seems not so surprising that, taking two different models but simulating them in a regime where the unusual features of one or the other models do not manifest (c.f. the response below to "why are you ignoring the weird parts of the Voronoi model"), one gets similar behavior.

Our reply: We are confused by the referee's argument. Our point is that on large enough length scales (typically above 5 cell radii, hence not very large), microscopic (cell-based), mesoscopic (normal mode based) and macroscopic (active continuum elasticity) approaches all nicely agree and, again, are not similar to thermal behaviour. We consider this as a validation of our approach. We don't see why it should be obvious from the start that different models have similar behaviours on large scale. We also emphasize again that active elastic theory is a new theory presented in this paper.

From our reply in the first round: Regarding the type of non-equilibrium driving: This is an intriguing question that is worth exploring but it is beyond the scope of this work. The advantage of the proposed active dynamics, i.e., uncorrelated random noise of the cells' directions, is that it is the only one that can be studied analytically in a relatively straightforward way. Even the simplest extension where each cell aligns its direction to its instantaneous velocity requires some strong simplifying assumptions to allow for analytical treatment. More involved models would only be treatable numerically.

Reviewer: Why is it beyond the scope of this work? If the work here has largely been done in a regime where the results are basically equilibrium-like (with the exception of, eg. parts of the distributions of velocities that the present work also does not try to explain), I do not see the deep novelty of this paper, since (as mentioned above), normal modes in thermal dense systems have been proposed as a source of the kind of quasi-localized excitations the authors call "swirl-like." If the non-equilibrium nature is, however, important, knowing whether the predictions here are made only for the simplest type of non-equilibrium dynamics seems important.

Our reply: The referee again assumes that our results are equilibrium-like, which they are not as explained above. So even in the simplest setting considered in this work, our results capture the non-equilibrium nature of the dynamics. Further effects like slight alignment interactions (strong alignment interactions are ruled out by the experimental observation of the absence of flocking) would certainly be perturbative. Including them at this stage may obscure the message rather than clarifying it.

Furthermore, including alignment interactions between cells in addition to the persistence would likely render the model analytically intractable. Given that is very hard to measure such interactions in experiments it would make comparison of simulations and experimental measurements prone to large uncertainties.

One could argue, though, that different sources of activity are possible, such as activated tension between cell junctions. Studying such effects, however, would require an entirely different set of models and experiments including fully segmented images of cells and is therefore very much beyond the scope of this work.

Reviewer: The abstract claims that uncorrelated, persistent motility that couples to the collective elastic modes of the cell sheet are sufficient to produce the characteristic “swirl-like” patterns seen in the experiments. Are the authors not concerned about Fig. 4A, which seems to show the biggest deviation between theory and experiment is for wavevectors that correspond to 1-10 cell radii – precisely the scale of the “swirl-like” patterns?

Our reply in the first round: Figure 4A indeed shows some discrepancy between the model and the experiment. Note however that it does show that above a length scale of 5-6 cell radii “the swirl size” theory and experiments agree with each other.

Reviewer’s response: I am deeply unimpressed by this answer. Yes, above the length scale of interest the system behaves elastically (as the authors mentioned in their response above, this is one of the theoretical ingredients in their description and is clearly true of the physical systems). But if all of the predictions are only quantitatively accurate (and quantitative is the authors’ own standard here) at the level of long-wavelength, simple elastic behavior, the authors have done a lot more work than they needed to in bothering with most of the details.

Our reply: We emphasize again that on long length scale, the dynamics is described by the active continuum elasticity theory we develop, and not by a “simple elastic behaviour” as stated by the referee.

From our reply in the first round: This is all that our mesoscale elastic model ever claimed. As to the origin of the discrepancy: We have so far excluded experimental imaging problems, local alignment between cells (see Figure 2 below) and finite size effects. We believe the issue is likely due to neither disks nor Voronoi cells being a particularly good model for the very local dynamics of (these) cells, due to, e.g., the absence of tangential friction terms. This is something for the community to look into! To further show that our model is actually a good fit for the data, we have added the real-space correlation functions and our new analytical predictions for it (eq. 14) in Figure 4C. Using the same fitting parameters as in Fig. 4B (Fourier space), it becomes clear that at scales above a couple of cell radii (11 microns), the model fits.

Reviewer: “The model fits” is doing a lot of work, here. Indeed, this is something of a summary of my whole impression of this paper. If Fig 4B and 4C I see curves that describe some data for some experiments, but certainly not all data for any of the experiments, and none of the data for some of the experiments. If part of the claims are about the applicability to multiple experiments/ cell lines / etc., why should I be convinced by these figures?

Our reply: We are confused by referee’s comment. Our theory and simulations are an excellent match to the experimental data at length scales beyond ~5 cell sizes, being able to fit the data with the smallest possible number of parameters (one). At length scales comparable to the cell size, our approach clearly fails. This is, however, not surprising as modelling that regime would require a far more detailed knowledge of cell-level properties. It is, therefore, not to be expected that this (or any simple) model would be able to fully fit any of

the data sets, and the discrepancy intriguingly suggests that current state-of-the art microscopic cell models are in fact not very good. Our study, however, shows that under a very simple assumption of self-propulsion coupled to elastic modes it is possible to quantitatively capture the mesoscale behaviour of a complex biological system. This is itself is, we argue, a remarkable result.

Reviewer #4

The authors have carefully answered my questions, including rewriting the manuscripts and performing additional controls/analysis, which have improved the manuscript. I recommend publication.

Our reply: We thank the reviewer for recommending our manuscript for publication.